# Resistance to Aflatoxin Accumulation in Maize Mediated by Host-Induced Silencing of the *Aspergillus flavus* Alkaline Protease (*alk*) Gene

**DOI:** 10.3390/jof7110904

**Published:** 2021-10-26

**Authors:** Olanike Omolehin, Yenjit Raruang, Dongfang Hu, Zhu-Qiang Han, Qijian Wei, Kan Wang, Kanniah Rajasekaran, Jeffrey W. Cary, Zhi-Yuan Chen

**Affiliations:** 1Department of Plant Pathology and Crop Physiology, Louisiana State University Agricultural Center, Baton Rouge, LA 70803, USA; oomolehin@agcenter.lsu.edu (O.O.); yraruang@agcenter.lsu.edu (Y.R.); dfhu2019@gmail.com (D.H.); 2Cash Crops Research Institute, Guangxi Academy of Agricultural Sciences, Nanning 530007, China; hanzhuqiang@163.com; 3Food and Feed Safety Research Unit, Southern Regional Research Center, United States Department of Agriculture—Agricultural Research Service, New Orleans, LA 70124, USA; qijian.wei@usda.gov (Q.W.); kanniah.rajasekaran@usda.gov (K.R.); jeff.cary@usda.gov (J.W.C.); 4Department of Agronomy, Iowa State University, Ames, IA 50011, USA; kanwang@iastate.edu

**Keywords:** *Aspergillus flavus*, host induced gene silencing, alkaline protease (*alk*), aflatoxin resistance, transgenic maize, small RNA, breeding, RNA interference

## Abstract

*Aspergillus flavus* is a fungal pathogen that infects maize and produces aflatoxins. Host-Induced Gene Silencing (HIGS) has been shown to reduce host infection by various fungal pathogens. Here, the *A. flavus* alkaline protease (*alk*) gene was targeted for silencing through HIGS. An RNAi vector carrying a portion of the *alk* gene was incorporated into the B104 maize genome. Four out of eight transformation events containing the *alk* gene, Alk-3, Alk-4, Alk-7 and Alk-9, were self-pollinated to T4/T6 generations. At T3, the Alk-transgenic lines showed up to 87% reduction in aflatoxin accumulation under laboratory conditions. T4 transgenic Alk-3 and Alk-7 lines, and T5 and T6 Alk-4 and Alk-9 showed an average of 84% reduction in aflatoxin accumulation compared to their null controls under field inoculations (*p* < 0.05). F_1_ hybrids of three elite maize inbred lines and the transgenic lines also showed significant improvement in aflatoxin resistance (*p* < 0.006 to *p* < 0.045). Reduced *A. flavus* growth and levels of fungal *ß-tubulin* DNA were observed in transgenic kernels during in vitro inoculation. Alk-4 transgenic leaf and immature kernel tissues also contained about 1000-fold higher levels of *alk*-specific small RNAs compared to null controls, indicating that the enhanced aflatoxin resistance in the transgenic maize kernels is due to suppression of *A. flavus* infection through HIGS of *alk* gene.

## 1. Introduction

*Aspergillus flavus* is an opportunistic plant pathogen that resides in the soil and has a global distribution. It reproduces predominately asexually through the release of a large number of conidia under natural conditions [1,2]. Major economically important crops including maize, peanut and cotton are highly susceptible to both *A. flavus* infection and subsequent contamination with carcinogenic secondary metabolites known as aflatoxins [3,4,5]. The annual economic loss of maize due to aflatoxin contamination was estimated to be in the hundreds of millions of dollars and has been predicted to reach as high as USD 1.68 billion due to global climate change [6], making aflatoxin contamination a serious economic threat to world trade of aflatoxin-susceptible crops. In addition to direct economic-related losses associated with aflatoxin contamination in maize, consumption of aflatoxin-contaminated maize products resulted in vulnerability to illnesses, hundreds of hospitalizations and deaths of humans and animals in various occasions across different countries [7,8,9,10].

The options available to manage aflatoxin contamination in maize include biological control through field application of atoxigenic *A. flavus* strains, such as NRRL30797, AF36 and NRRL21882, which are capable of reducing aflatoxin accumulation up to 94.8% in maize by competitively displacing native *A. flavus* strains that produce aflatoxin [11,12,13]. However, the adoption of biological control agents relies on resource availability among growers. In addition, the efficacy of biological control strains is affected by abiotic factors, which often lead to questions on the resilience and stability of these biological control agents under diverse environmental conditions [14,15]. Therefore, alternative strategies that provide a consistent protection of maize against *A. flavus* infection and aflatoxin contamination without extra costs, such as seed-based host resistance, are highly desired.

Plant microRNAs (miRNA), a type of small RNAs (21–22 nucleotides), have been found in several studies to play important role in crop resistance by interfering with pathogen gene expression through a natural phenomenon known as RNA interference (RNAi) or gene silencing [16,17,18], by triggering the cleavage of target transcripts or repressing translation [19,20]. Some recent studies further demonstrate that small RNA communication is bidirectional and that small RNAs produced by pathogens can also suppress host immune/defense responses during the pathogen infection [21,22,23]. For example, *Botrytis cinerea* was able to deliver small RNAs into *Arabidopsis* and tomato to suppress the expression of their resistance genes [22]. Along the same idea, small RNAs of pathogen origin expressed in the hosts have been successfully employed to enhance crop resistance to various diseases [24,25,26,27]. This approach is called Host-Induced Gene Silencing (HIGS). HIGS was first reported when an effector gene of root-knot nematode was suppressed by expressing dsRNA in an *Arabidopsis* host [28]. HIGS has since been used to enhance host resistance to pathogens by targeting genes important for virulence or infection in HIGS vectors designed to initiate specific disruption of pathogen mRNA during infection of plants [26,29,30]. For example, reduced wheat leaf rust infection was observed in wheat by targeting three pathogenicity genes that encode a *Pt* MAP-kinase (*PtMAPK1*), a cyclophilin (*PtCYC1*) and a calcineurin B (*PtCNB1*) [29].

HIGS has also been shown effective in enhancing maize resistance to *A. flavus* infection/aflatoxin contamination. Masanga et al. [31] transformed maize with a hairpin construct targeting the *A. flavus* transcription factor, *aflR*, and reported a 14-fold reduction in aflatoxin accumulation in the transgenic lines under greenhouse conditions but also reported off target effect of RNAi that caused undesirable phenotypic appearance of the newly generated transgenic lines. In a separate study, an RNAi-cassette containing the *A. flavus aflC* gene was selected as a target for silencing by Thakare et al. [32]. Another study silenced the amylase-1 gene (*amy1*) of *A. flavus* and observed that reduced expression of the *amy1* gene correlated with decrease in fungal growth and reduced level of aflatoxin under laboratory condition. However, none of these three studies evaluated their transgenic lines under field conditions. A recent study that targeted the aflatoxin biosynthesis pathway gene *aflM* for silencing successfully reduced aflatoxin accumulation up to 76% under three consecutive years of field evaluations [33]. This study clearly demonstrated the full potential of HIGS as a reliable and sustainable approach to manage aflatoxin accumulation in maize by silencing the *A. flavus aflM* gene.

The objective of this study was to target a different gene that is related to *A. flavus* virulence instead of aflatoxin biosynthesis for silencing to reduce the infection and subsequent aflatoxin production. The gene encoding the alkaline protease (*alk*) protein was selected based on its role previously reported in Aspergilli. The *alk* gene was initially cloned in *A. nidulans* and *A. oryzae* [34,35]. Chen et al. [36] clearly showed that a 33 kDa protein with a 95% homology with the *A. flavus* alkaline protease was produced at a higher level during infection of maize embryos compared to the control.

In order to determine if silencing *alk* in *A. flavus* has a potential to enhance maize resistance to *A. flavus*, a stable transformation of a maize line (B104) with a HIGS construct designed to carry a partial fragment of the *alk* gene for silencing was carried out. Of the eight transformants, four were characterized for the presence of the target gene RNAi cassette (Alk-RNAi). Evaluation of these four events under artificial *A. flavus* inoculation reduced up to 87% and 84% aflatoxin accumulation in the transgenic lines, under laboratory and field conditions, respectively. No undesirable phenotypic changes were observed when compared to their null controls under field condition. In addition, reduced *A. flavus* biomass was observed in transgenic kernels compared to their null controls based on PCR quantification of *A. flavus*
*β-tubulin* DNA. F1 crosses between the homozygous transgenic Alk-4 and Alk-9 lines with elite inbred lines PHN46, LH195 and PHG39 also showed enhanced aflatoxin resistance. Further investigation through RNA sequencing detected significantly higher levels of gene specific small interfering RNAs comprising of sense and anti-sense strands of 21–24 nucleotides in the transgenic leaf and kernel tissues. These small RNAs may have served as precursors for silencing the *alk* gene thereby reducing the infection and aflatoxin accumulation. This is the first reported study in targeting *A. flavus* alkaline protease gene for aflatoxin management with an emphasis on field evaluation and the potential development of aflatoxin resistant transgenic maize lines in elite backgrounds.

## 2. Materials and Methods

### 2.1. Construction of Alk-RNAi Vector

An alkaline protease gene (*alk*) from *A. flavus* AF13 (GenBank accession number AF324246) was selected as the target for suppression. The construct used in this study was a Gateway-based vector constructed in a similar manner as previously reported [37]. The 5′ and 3′ arms of the *alk* coding region were PCR amplified using specific primers (Forward GCG TTA CCG TTG TAG GCA AG and Reverse TCC AGA AGA GCA ACA ACC GC) (Appendix A) and cloned into pDONR P4-P1R and pDONR P2RP3 (Invitrogen, Carlsbad, CA, USA), respectively, using BP clonase as previously described [37]. The derived vectors, namely, pENTR-L4-5′alk-R1 and pENTR-R2-3′alk-L3 were confirmed by sequencing. The pDONR221-PR10-intron-CmR containing a *PR10* intron was constructed previously [37]. pENTR-L4-5′alk-R1, pENTR-R2-3′alk-L3 and the pDONR221-*PR10* intron-CmR vectors were combined with the pBS-d35S-R4-R3 vector to produce pBS-d35S-attB4-5′alk-attB1-PR10 intron-CmR-attB2-3′alk-attB3 vector using LR clonase (Invitrogen, Carlsbad, CA, United States). EcoRI and SacI were used to digest the vector and the resulting d35S-attB4-5′alk-attB1-PR10 intron-CmR-attB2-3′alk-attB3 cassette was cloned into pTF102 at the corresponding restriction sites [38]. The resulting maize RNAi transformation vector (pTF102-d35S-alk RNAi vector) (Appendix A) was sequenced to verify the assembly.

### 2.2. Maize Transformation with Alk-RNA Interference Vector and Regeneration

The final Alk-RNAi construct described above was transformed into *Agrobacterium* strain EHA101 and then into immature B104 maize embryos. Transformation was carried out at the Plant Transformation Facility (PTF), Iowa State University (ISU) following the protocol of maize transformation as detailed by Raji et al. [39]. The resulting transgenic plants regenerated from stable calli were recovered using the bialaphos-containing media. Plants regenerated from transgenic events were pollinated using B104 pollen and harvested at maturity between April and June 2013.

### 2.3. Confirmation of Transformation and Target Gene Expression

An initial confirmation of target *alk* gene and its expression in all the transformation events was carried out using genomic DNA and RNA isolated from T0 leaf tissues. DNA was extracted using a modified CTAB method as described by Doyle and Doyle [40]. The quality and quantity of isolated total DNA was determined using a Nano-Drop ND-1000 Spectrophotometer (Thermo Scientific, Wilmington, DE, USA). DNA was diluted to 50 ng/µL and used as the template for PCR with specific primers corresponding to the *alk* gene (Forward primer: GCG TTA CCG TTG TAG GCA AG and Reverse primer: TCC AGA AGA GCA ACA ACC GC) (Appendix A). Conventional PCR was conducted in a 20 μL volume containing 0.2 µL of Taq polymerase (5 unit/µL), 2 µL 1xdNTP (1 mM), 2 µL of 10× PCR buffer and 1 µL each primer (0.4 μM). The PCR was performed under the following conditions: initialization was at 95 °C for 3 min, followed by denaturation at 95 °C for 50 s, annealing at 56.4 °C for 60 s, elongation for 1 min 30 s for 35 cycles with a final elongation at 72 °C for 10 min and followed by holding at 12 °C. The plasmid containing the Alk-RNAi construct was used as the positive control while DNA extracted from B104 was used as a negative control. The PCR products were visualized using 1% agarose gel after mixing with 3.3 µL of 6× DNA loading dye (containing bromophenol blue and xylene cyanol FF). The expression level of target gene in young transgenic T0 leaves was determined using qRT-PCR with cDNA from RNA isolated from leaf tissues as a template and the specific *alk* primers: forward primer, CTCGCTGCCCTTGAGAAC and reverse primer, GCTGCCCTTAACATCCTTGA, and a TaqMan probe, FAM/CAA CTC CTT/ZEN/GAT GCG CTT GGTCAC/3IABkFQ (Appendix A). The qRT-PCR was performed using Bio-Rad CFX Connect TM Real-Time System (Bio-Rad, Hercules, CA, USA) under the following conditions: initial denaturing at 95 °C for 3 min, followed by 40 cycles of denaturing at 95 °C for 15 s, annealing and elongating at 55 °C for 30 s. The expression level of 18S rRNA was used as an internal control to normalize the level of target gene expression. The amplification efficiency of each primer pair used in this study was determined through serial dilutions and was taken into account in calculating target gene expression. Four of the positive transformants, namely, Alk-3, Alk-4, Alk-7 and Alk-9 were selected for further evaluation studies.

### 2.4. Genotyping, Zygosity and Transgene Copy Number Assessment Using Real-Time and Droplet Digital PCR

Conventional PCR analysis was done to determine the segregation for the *alk* gene at T2. Maize lines transformed with Alk-RNAi construct but are absent for *alk* gene are considered non-transgenic controls or “null” line for the same event and referred to as negative control alongside the B104 (transformation background line) for all field and laboratory studies. Those segregating lines containing the *alk* gene were heterozygous/homozygous transgenic lines. These lines were self-pollinated for several generations in order to attain homozygosity for the *alk* gene. qRT-PCR was conducted to verify the zygosity of the *alk*-positive, segregating maize lines after each selfing. Zygosity test was performed using the Bio-Rad CFX Connect ^TM^ Real-Time System in a final volume of 25 µL containing 1× TaqMan Universal PCR Master Mix, 200 nM of each primer, 100 nM of probe (Appendix A) and 150 ng of genomic DNA from leaf tissue under the following conditions: 95 °C for 3 min, 40 amplification cycles of 95 °C for 15 s and annealing and amplifying at 55 °C for 60 s. Three technical replicates were included for each sample. The alcohol dehydrogenase (*adh1*) gene (a single copy gene in maize) was used as a normalizer. The *alk/adh* ratio in the leaf tissue of T0 transgenic plants, which is expected to be heterozygous for the target gene (*alk*), was used as the basis of comparison to determine the zygosity. Zygosity was calculated using an equation based on the threshold cycle (*Ct*) values of the *alk* gene and the *adh1* normalizer: Z = 2^[T4 *Ct* (*adh*) − T4 *Ct* (*alk*)] − [T0 *Ct* (*adh*) − T0 *Ct* (*alk*)]^. The primers for *alk* are expected to detect both 5′ and 3′ arms of the *alk* in the transgenic lines, and therefore, the expected Z value for the T0 transgenic lines, which are heterozygous, is 1. Samples doubling the value of Z calculated in the heterozygous were considered homozygous for the target gene.

In addition, DNA isolated from the following leaf tissues of events, Alk-3 (T0), Alk-4 (T0), Alk-7 (T0), Alk-9 (T0), Alk-3 (T4), Alk-4 (T4), Alk-7 (T4) and Alk-9 (T4), were sent to the Interdisciplinary Center for Biotechnology Research (ICBR), University of Florida (Gainesville, FL, USA), for precise assessment of transgene copy number using droplet digital PCR (ddPCR) [41,42,43,44]. ddPCR reactions were prepared using 100 nM of forward and reverse primers specific to *alk* transgene and maize *adh1* gene (Appendix A), 12.5 μL of a Bio Rad ddPCR Supermix including an intercalating fluorescent dye, polymerase, Mg^2+^ and dNTPs and MilliQ autoclaved water in a total reaction of volume of 25 μL. Droplet mix was cycled through a PCR program (denaturing at 95 °C for 10 min, followed by 40 cycles of 94 °C 30 s, 56 °C 30 s and 60 °C 30 s, with a final inactivation at 98 °C for 10 min before holding the reactions at 4 °C) using a deep-well thermal cycler (C1000 Touch, 185-1196, Bio-Rad, Hercules, CA, USA). The reactions were immediately analyzed in a droplet reader (QX200, 186-4003, Bio-Rad, Hercules, CA, USA).

### 2.5. Seed Increase of Selected Transgenic Lines by Self-Pollination and Generation of Crosses with Elite Maize Lines

Four cycles of self-pollination were carried out using the Alk-RNAi T2 lines between 2016 and 2019 to ensure homozygosity for the *alk* target gene in the transgenic maize lines. T2 seeds were increased to T3 in the spring of 2016 and from T3 to T6 from 2017 to 2019 in the Burden Farm and Museum in Baton Rouge, Louisiana. In addition, pollen grains from homozygous Alk-4 and null-4 plants were used to pollinate the silk of the LH195, PHN46, and PHG39 elite inbred lines to incorporate the *alk* transgene into elite maize backgrounds to produce F1 hybrids in 2018. In 2019, pollens from homozygous transgenic Alk-9 and null-9 lines were used to cross with the above elite lines to produce F1 hybrids. The information on the number of seeds used, seedlings planted, plants pollinated and ears inoculated and harvested from T3 to T6 and for the production of F1 crosses was summarized in Table 1. Phenotypic evaluation of maize lines was carried out by selecting 10 to 12 representative maize plants at the reproductive growth stage. Plant height was measured from the base of the plant to the flag leaves (cm). Dates at 50% tasseling and silking were recorded for selected plants (number of days). After harvest, the cob length was measured from the tip to the base. After shelling the kernels, total number of kernels from each cob and weight (g) of 100 kernels were recorded.

### 2.6. Evaluation of Transgenic Maize Lines for Changes in Aflatoxin Resistance under Laboratory Assay and Field Inoculation Conditions

T3 generation homozygous kernels from Alk-3, Alk-4, Alk-7 and Alk-9 events were evaluated for their changes in resistance to aflatoxin in comparison to B104 and non-transgenic null kernels using the Kernel Screening Assay (KSA) as described by Brown et al. [45]. Twenty kernels from each transformation event and non-transgenic controls (null) were first surface-sterilized in 70% ethanol for 5 min with constant stirring and rinsed with distilled water for 5 min for three times and allowed to dry. Sterilized and dried seeds were inoculated by dipping in newly prepared *A. flavus* inoculum (10 mL of 4 × 10^6^ conidia/mL) that was prepared using toxigenic *A. flavus* AF13 (ATCC 96044, SRRC 1273) strain maintained on V8 medium (5% V8 juice, 2% agar and pH 5.2). Spores were freshly harvested from 7-day-old PDA plates grown at 30 °C and suspended in 0.01% SDS solution. The initial concentration was determined with a hemocytometer and diluted to 4 × 10^6^ conidia/mL. Each inoculated seed was placed in a 35 × 10 mm petri-dish and incubated at 30 °C under 100% humidity in the dark for seven days, after which kernels were dried at 65 °C for 72 h to stop *A. flavus* growth. Dried individual kernels were then pulverized to powder using a coffee blender (Mr. Coffee^®^, Boca Raton, FL, USA).

For field evaluation of aflatoxin resistance from T4 (2017) to T6 (2019), 7–15 immature ears from each line were inoculated 14 days after pollination at four points in the middle of the ear with approximately 0.5 mL/site of the *A. flavus* AF13 inoculum using the side-needle method with an Indico tree-marking gun (Forestry Suppliers, Jackson, MS, United States) and a 15-gauge hypodermic needle as described by Williams et al. [46]. Inoculated ears were harvested after maturity (about 60 days after pollination). Four kernels surrounding each inoculation point were retrieved and ground using a coffee blender (Mr. Coffee^®^, Boca Raton, FL, USA) as one sample. In 2017, the aflatoxin production analysis was carried out using 48 replicates (12 ears) of Alk-4 homozygous, 44 (12 ears) of Alk-4 null and 20 (5 ears) of the B104 control and 36 (9 ears) of Alk-9 homozygous, 27 (7 ears) of Alk-9 null. In 2018, two lines each from the Alk-4 (B4-33 and B4-37) and Alk-9 (B9-15 and B9-19) events were re-evaluated under field inoculation conditions. The inoculum concentration that was used in the 2017 field study was 4 × 10^6^ conidia/mL, which was reduced to 1 × 10^5^ conidia/mL in the following two years to better simulate natural infection.

The homozygous lines of Alk-4 and Alk-9 events that showed lower aflatoxin accumulation among the two transgenic lines used in this study (i.e., B4-37 and B9-19) were evaluated again under field conditions in 2019 using kernels from T6 generation with the same inoculation treatment as in 2018. In addition, kernels from two additional events, Alk-3 and Alk-7, at T4 generation were evaluated under the same field conditions. Three to four replicated samples per ear with each containing four intact kernels that surrounded each inoculation site with 10–14 inoculated ears per events were analyzed for changes in aflatoxin levels.

For evaluation of aflatoxin resistance in F_1_ crosses, 9–15 immature ears were inoculated two weeks after cross-pollination as above; however, kernels from the lower half of the mature ears were collected, ground into powder using a GRINDOMIX knife mill GM 200 milling machine (Retsch USA, Newtown, PA, USA) at 50 Hz speed for 10 s per sample under room temperature, and three subsamples of 2 g each were used for aflatoxin extractions as described by Sobolev and Dorner [47], and the levels of aflatoxin accumulation were analyzed using the High-Performance Liquid Chromatography (HPLC) as described below.

### 2.7. Aflatoxin Extraction and Quantification

Ground kernel samples were weighed before aflatoxin extraction using 20 mL of 80% methanol at room temperature in 50 mL flasks under continuous shaking at 60 rpm for 1 h. The extract was then filtered through a funnel with 100 mm No. 1 Whatman filter paper into a 50 mL glass beaker. The initial stock filtrate (100 µL) from each sample was diluted 10-fold with 80% methanol in a 1.5 mL tube before filtering through a 1.5 mL alumina-basic column containing about 0.5 g of alumina silica (CAT 1344-28-1, Fisher Chemical, Switzerland) [47]. The final filtrate was collected into properly labelled vials for aflatoxin quantification.

Aflatoxin was quantified using a reversed-phase HPLC as described by Sweany et al. [48]. A Waters e2695 HPLC (Waters Corporation, Milford, MA, USA), and the accompanying Empower software was used for the detection and analysis of aflatoxin B1 peak. Each extracted aflatoxin sample (10 μL) in methanol was then injected into HPLC using an auto sampler and separated with a Nova-Pak C18 4 µm 3.9 × 150 mm column at 38 °C. The mobile phase was 37.5% methanol and 62.5% water at a 0.8 mL/min flow rate. Each sample was run for 16 min with the B1 peak emerging at approximately 13.5 min. Aflatoxin detection was achieved using an in-line post-column derivatization by engaging a UV light in a Photochemical Reactor for Enhanced Detection (Aura Industries Inc., New York, NY, USA). This was followed by an excitation at 365 nm wavelength and detection of emission at 440 nm with a Waters 2475 FLR Detector (Waters Corp., Milford, MA, USA) [49]. Aflatoxin standard obtained from Sigma Aldrich (St. Louis, MO, USA) was serially diluted and used for the standard curve construction.

### 2.8. Evaluation of Aspergillus flavus Biomass in Maize Kernels

Mature transgenic and non-transgenic (null) maize kernels were inoculated and incubated as described above for the KSA. After 72 h of inoculation, kernels were photographed and observed for *A. flavus* infection under a dissecting scope. Kernels were then dried at 65 °C for four hours to stop *A. flavus* growth before grinding into powder for genomic DNA isolation. DNA were diluted to 100 ng/µL and used as template for PCR with primer pairs targeting the *A. flavus β-tubulin* and maize membrane protein (MEP) genes (Appendix A). DNA from inoculated and non-inoculated B104 control kernels were used as controls. PCR analyses were carried out under the following conditions: 95 °C for 3 min; 35 cycles of 95 °C for 15 s, 64 °C for 30 s, 72 °C for 60 s for the MEP primers and 62 °C for annealing of the β-tub primers. The amount of fungal DNA was calculated by the Ct value for β-tub divided by the amount of maize kernel DNA (as calculated by the MEP Ct values). The ∆∆Cq method was used to calculate the normalized β-tubulin levels relative to the maize MEP gene, with its level in B104 control set as 1.

### 2.9. Total RNA Isolation, Small RNA Library Construction, Sequencing and Bioinformatics the Detection of Alk-specific Small RNA

Total RNAs were isolated from T3 leaf tissues of Alk-4 and the B104 non-transgenic line. At T4, total RNAs were also isolated from immature maize kernel tissues of Alk-4 homozygous and their null controls 14 days after self-pollination. After grinding kernel or leaf tissue to a fine powder, total RNA was extracted from 200 mg of kernel or leaf tissue using the TRIzol reagent according to the manufacturer’s instructions and then cleaned with RNeasy Plant Mini Kit (QIAGEN, Hilden, Germany). The quality and quantity of isolated total RNAs was determined using a Nanodrop. Indexed sRNA libraries were constructed from the enriched sRNA fractions with the TruSeq Small RNA Library Preparation Kits (RS-200-0012, Illumina, San Diego, CA, USA) according to the manufacturer’s instructions. Indexed sRNA libraries were sequenced on the Illumina HiSeq 2500 platform at the Genomic Science Laboratory of the NC State University (Raleigh, NC, USA) in 2016 (T3 leaf tissue) and on Illumina HiSeq 4000 at the Genomic Sequencing Core at UC Davis (Davis, CA, USA) in 2017 (T4 kernels), respectively. The adapters and indexes were trimmed using Cutadapt [50] version 1.12, and the reads were mapped to the maize and *A. flavus* genome sequences using Bowtie2 [51] version 2.1.0. to identify sRNAs with a perfect match. The Awk command lines were used to extract small RNA specific to the targeted *alk* gene. R [52] was used to generate read counts. The sRNA sequence locus position and read length was illustrated using Microsoft Excel (Microsoft Corp., Seattle, WA, USA).

### 2.10. Statistical Analyses

Statistical analysis was conducted using SAS version 9.4 (Statistical Analysis System, SAS Institute, Cary, NC, USA). Analyses of variance (ANOVAs) were calculated using Proc Mixed. Post hoc comparison of means was calculated using Turkey’s LSD means [53]. Significance in this study was defined by a confidence interval ≥95% (α = 0.05).

## 3. Results

### 3.1. Construction and Transformation of HIGS Vector into Maize and Confirmation of alk Target Gene

The HIGS vector was constructed as shown in Appendix A. After inserting the alk 5′ and 3′ arm and the intron containing chloramphenicol selection marker (CmR) into pBS-d35S-attR4- attR4 through LR recombination, the resulting construct was verified through digestions with EcoR V and EcoR I/Kpn I restriction enzymes, and the resulting fragment sizes were in agreement with expected sizes. The T-DNA portion of the above vector was then excised with BamH I and Sac I, which was inserted into the corresponding site of pTF102. The final construct (Appendix A) with inverted repeats of the *alk* fragment was verified through digestions with EcoR V and Kpn I restriction enzymes (Appendix A). The fragment sizes were in agreement with the expected 0.2, 1.7 and 8.9 kb and 0.2, 1.1, 1.3 and 8.7 kb sizes (Appendix A). This construct was capable of producing a 290-bp *alk* dsRNA transcript in the host plant.

The construct was transformed into immature embryo of maize B104 line using the Agrobacterium infection method. Among twelve independent transformation events, eight were confirmed positive, and four (Alk-1, 5, 8 and 12) were negative for the alk target gene when the genomic DNA from the T0 leaf tissue was analyzed through PCR (Figure 1A). Two of the positive transformation events (Alk-7 and Alk-10) showed the highest target gene expression and the negative events (such as Alk-1, Alk-8 and Alk-12) showed no detectable level of target gene expression as in B104 control (Figure 1B).

### 3.2. Variation in Gene Copy Number and Zygosity

The number of *alk* gene integration was determined based on PCR genotyping of seedlings generated from T3 generation kernels using chi-square analysis. The number of integration was estimated based on the probability of chi-square value Χ^2^ = ∑ (observed-expected)^2^/(expected) exceeding the critical value (3.841, *p* < 0.05) to either reject or accept the null hypothesis of being one or two integrations. Segregation of seedlings with transgene and without (null) is expected to be 3:1 (transgene: null) for a single integration. The results indicated a single integration of *alk* gene for the four events under study (Alk-3, Alk-4, Alk-7 and Alk-9) (Table 2). All four events were further quantified for *alk* copy number using the droplet digital PCR (ddPCR). The ratio of gene copy number of *alk/adh1* for genomic DNA samples from T0 and T4 generation leaves revealed the presence of a single copy of *alk* for Alk-3, 4, 6 and 11 and two copies of the transgene in Alk-7, 9 and 10 events (Table 3).

### 3.3. Aflatoxin Production in T3 kernels of Alk-3, Alk-4, Alk-7 and Alk-9 Events under Laboratory Kernel Screening Assay (KSA) Conditions

The KSA assay was used to assess the aflatoxin resistance of T3 transgenic kernels. A significant reduction in aflatoxin production was observed in the kernels of homozygous lines of Alk-3, Alk-4, Alk-7 and Alk-9 events compared with their null controls (*p* < 0.02, *p* < 0.016, *p* < 0.003 and *p* < 0.042, respectively) (Figure 2). Reduction in aflatoxin levels achieved under in vitro conditions varied among the transgenic lines. The highest percentage of reduction in aflatoxin accumulation was observed in Alk-7 transgenic with 87% lower aflatoxin production compared to its null control. Alk-9 transgenic had the lowest percent reduction with 65% decrease in aflatoxin accumulation compared to the null.

### 3.4. Phenotypic Assessment of Alk-RNAi Homozygous Transgenic and Null Control Plants

Ten representative T4 plants of the selected lines grown in the field were evaluated for phenotypic performance. Plant height at flowering ranged from 132 to 142 cm across transgenic Alk-RNAi homozygous lines and the null controls (Table 4). Average ear length ranging from 13 to 14.9 cm was observed for homozygous lines and the null controls (Table 4). Likewise, the 100-kernel weight was similar for homozygous lines and the null controls. However, there was a significant difference in plant height between transgenic and null for the Alk-9 event. Delayed tasseling and shorter plant height were observed in the transgenic plants of Alk-9. One hundred kernel weight of the Alk-9 homozygous line was also significantly lower than that of the null control (Table 4). However, this difference is not always associated with the presence of the transgene, such as in the case of Alk-3, which had significantly higher 100-kernel weight than the null. The variation in plant height, days to tasseling, ear length and kernel weight did not significantly affect the morphology of the Alk-9 transgenic line (Appendix A). These field evaluations suggest that there are no clear detrimental morphological changes that may be associated with the presence of the Alk-RNAi silencing construct or its possible off-target effect in the selected lines.

### 3.5. Aflatoxin Production of Alk-4 and Alk-9 Events at T4 to T6 Generations and Alk-3 and Alk-7 at T4 Generation under Field Inoculation Conditions

For aflatoxin production in T4 generation kernels of Alk-4 and Alk-9 conducted in 2017, only a border-line significant difference was observed between Alk-4 homozygous line and its null control (*p* < 0.058) but not between the Alk-9 homozygous line and its null control (*p* < 0.27) (Appendix A) due to the presence of severe *Fusarium spp*. infection of maize ears in the field, which greatly reduced the overall aflatoxin production. In 2018, two lines from each of the T5 generation Alk-4 and Alk-9 transgenic events were evaluated through field inoculation, and all showed significantly lower aflatoxin production ranging from 66–73% and 80–96% compared to their null and B104 controls (*p* < 0.0017 to *p* < 0.003) (Figure 3A). In 2019, significantly lower aflatoxin production was again observed in the homozygous transgenic kernels of these two events (Alk-4 and Alk-9) at T6 compared with null controls (*p* < 0.04, 0.002, respectively) (Figure 3B). Aflatoxin production was reduced by 73–83% in the homozygous transgenic lines compared to their null controls. Moreover, in 2019, two more events (Alk-3 and Alk-7) of T4 generation kernels were evaluated, and they showed up to 83% reduction in aflatoxin (*p* < 0.032 and 0.006, respectively) (Figure 3C).

### 3.6. Reduced Aflatoxin Accumulation in Field Inoculated F1 Crosses between Homozygous T5 Generation of Alk-4 and T6 Generation of Alk-9 with Three Elite Inbred Lines

In order to determine whether the enhanced aflatoxin resistance observed in the homozygous transgenic lines was due to the presence of *alk* transgene, the homozygous Alk-4 and Alk-9 lines were crossed with three elite inbred lines. Significantly less aflatoxin production was observed in the kernels of F1 crosses with the presence of the transgene than that in the control kernels without the transgene (Figure 4) in all three different crosses (*p* < 0.006 to *p* < 0.045). The Alk-4 F1 and Alk-9 F1 hybrids accumulated 67% to 89%, 66% to 68% and 61% to 86% less aflatoxin compared to the elite lines LH195, PHN46 and PHG39, respectively. On an average, 67% (for crosses with PHN46) to 78% (for crosses with LH195) reduction in aflatoxin accumulation was achieved in the hybrid lines. These results demonstrate that the presence of the transgene is the cause of reduced aflatoxin production in the F1 crosses between the homozygous Alk-4 or Alk-9 line and the elite inbred lines.

### 3.7. A. flavus Growth in Alk Lines and Null Controls

A separate study was conducted to determine whether the presence of the transgene in the kernels affects the *A. flavus* infection/growth. Although the fungus was visible on both transgenic and null kernels at 72 hpi, the rate of *A. flavus* growth in the transgenic kernels for the four Alk events was slower compared to the fungal growth on the null and B104 kernels based on visual observation (Figure 5A). Differential *A. flavus* growth rate in the transgenic kernels was confirmed using qRT-PCR. The inoculated transgenic kernels contained significantly lower levels of *A. flavus β-tub* DNA than those in the inoculated null and B104 controls (Figure 5B). Reduction in *β-tub* DNA ranged from 60 to 85% in the transgenic kernels compared to null control kernels.

### 3.8. Small RNA Production in Alk-RNAi Homozygous Transgenic and Null Controls

In order to determine whether the enhanced aflatoxin resistance observed in the Alk-RNAi homozygous transgenic T4 kernels was due to the presence of *alk*-specific small RNA produced from the introduced Alk-RNAi cassette, small RNAs from transgenic and null leaves and kernel tissues were sequenced and analyzed (Table 5). When the small RNA libraries from the leaf tissue of homozygous Alk-4 and B104 lines were sequenced, 13,699 and 6238 small RNA reads were aligned to *A. flavus* genome. Among them, 9574 (70%) and 7 (0.11%) reads were specifically aligned to *alk*, respectively. The total number of small RNA reads aligned to *A. flavus* was 52,476 in Alk-4 homozygous kernel tissue and 64,291 in the null line, respectively. Among them, 6606 read counts (11%) and 8 read counts (0.12‰) of small RNA from Alk-4 homozygous and null kernel tissues were specifically aligned to *alk*, respectively. The alignment of the small RNA sequences from homozygous transgenic plants to other random non-target *A. flavus* genes yielded a similar number of reads of small RNAs to that of the null and B104 control plants that were aligned to alk (Table 5). The most abundant small RNAs were 21, 22 and 24 bp in length in the transgenic lines (Figure 6A). In addition, both *alk*-specific anti-sense and sense small RNAs, which are the critical components of gene-specific RNA silencing, were observed. The majority of the sequenced small RNAs were from the antisense strand (Figure 6B). The small RNA profiling also showed high read counts in the 290 bp region that coincides with the size of the *alk* transgene inserted into the transgenic lines (Figure 6C).

## 4. Discussion

The overall goal of this study was to determine if targeting the alkaline protease (*alk*) gene in the filamentous *A. flavus* pathogen using HIGS could reduce its infection of maize, and subsequent aflatoxin contamination. The decision to target *alk* gene for suppression was based on a previous study that showed the *ALK* protein was dominantly produced in *A. flavus* infected maize kernels and the reduction of *alk* by phenylmethylsulfonyl fluoride led to significantly lower aflatoxin production in specific growth medium [36]. The present study hypothesized that since *alk* plays a significant role in *A. flavus* virulence, silencing the gene using HIGS may have a great potential in enhancing maize resistance to *A. flavus* infection and aflatoxin accumulation.

Out of the 12 independent events produced, eight were positive for containing the *alk* gene. During evaluation, the T1 transgenic kernels were observed to have lower levels of aflatoxin accumulation in comparison to the non-transgenic kernels (Appendix A). After seed increase through self-pollination to T3 generation, the kernels of four selected events showed significantly lower aflatoxin accumulation compared to the kernels that segregated for no *alk* gene (null) under artificial inoculation in the laboratory KSA assay (Figure 2). Their enhanced aflatoxin resistance was further verified under artificial inoculations in field trials from 2017 to 2019 (kernels of T4, T5 and T6 Alk-4 and Alk-9 and T4 Alk-3 and Alk-7) with a reduction in aflatoxin production as high as 96%. The results from these repeated evaluations of multiple Alk transformation lines at different generations demonstrated that the introduced RNAi vector targeting the *A. flavus alk* gene resulted in the observed enhanced aflatoxin resistance in the transgenic maize lines (Figure 3). These maize lines could be useful germplasms for breeding commercial maize varieties with aflatoxin resistance. In an attempt to determine the usefulness of these transgenic lines in developing aflatoxin-resistant commercial maize lines, Alk-4 and Alk-9 transgenic plants were crossed with three elite inbred maize lines in 2018 and 2019, respectively. The resulting hybrids also showed significantly reduced aflatoxin accumulation confirming that the enhanced aflatoxin resistance in the newly developed transgenic lines is due to the presence of the *alk* transgene (Figure 4). Some of these lines will be further evaluated at different locations to determine the reliability of their aflatoxin resistance status and yield potentials.

Furthermore, a significant reduction in fungal biomass in the transgenic kernels at 72 hpi compared with null kernels was observed under laboratory conditions (Figure 5), confirming that *alk* plays a major role in *A. flavus* infection of maize since the *β-tub* gene is a key constitutive gene in fungi [54], and its level of DNA correlates to the level of *A. flavus* growth in the kernels, thereby suggesting that the level of *A. flavus* biomass in maize kernels was significantly higher in null compared to the transgenic kernels. This result suggests that the reduction in the overall aflatoxin accumulation may be attributed to the reduction in *A. flavus* infection and growth in the maize kernels. In order to verify that our observed reduction in aflatoxin and lower growth of *A. flavus* in the transgenic maize was due to the RNAi construct introduced through transformation, small RNA libraries were constructed from kernel and leaf tissues of Alk-4 transgenic and the null controls and sequenced. The presence of significantly higher levels of the *alk*-specific sense and anti-sense strands of small RNAs in the transgenic compared to the null and B104 controls were also observed (Figure 6; Table 5). In line with previous studies on the role of 22 nt sRNAs in transitive and systemic spread of RNAi [55,56,57], significantly high amounts of the 22 nt long sRNA were observed in the transgenic Alk-4 line (Figure 6A). It may be interesting to examine the levels of 22 nt sRNA in different events and determine whether they correlate with changes in aflatoxin resistance in future studies.

Varying levels of aflatoxin reduction in the transgenic lines compared to null lines from different events were observed (Figure 3). This variation may be due to difference in target gene copy numbers among different events. The influence of location of integration and copy number on transgene expression is well documented in earlier transgenic studies [58,59,60]. In these studies, integration and copy number of the transgene were found to have either positive or non-significant influence on resistance to virus in transgenic lines. Several other studies also found that multiple transgene copy numbers improved the resistance [61,62,63]. Based on our analyses through digital droplet PCR (ddPCR), conventional PCR and chi-square evaluation, the four events used in the present study had gene copies ranging from one to two but segregated as a single integration of the target gene since the transgenic lines followed the Mendelian segregation pattern of a single gene. The presence of two copies of the transgene leads to another concern: impact on phenotypical changes and non-target gene expression. Similar plant height, kernel weight, flowering time and pollen shed were observed between transgenic and non-transgenic null controls with the exception of few cases that may be attributed to soil variations in parts of the field where the different lines were grown, rather than genetic factors of the lines. The transgenic Alk-9 line showed delay in flowering time compared to the null control; however, the resulting transgenic ears and kernel sets were not negatively affected (Table 3; Appendix A). In addition, sustained aflatoxin resistance was observed in subsequent evaluation of the Alk-9 in T5 and T6 generations (Figure 3).

Another point worthy of mention is that although the levels of aflatoxin accumulation inside kernels vary greatly depending on the inoculum concentration used, the method of inoculation and the test conditions, the percentage of aflatoxin reduction in the transgenic lines compared to their null controls was fairly consistent across the board. Under the laboratory assay, kernels were dipped into inoculum, thereby allowing for penetrations of fungal inoculum and the initiation of infection through various kernel parts. In the field, the inoculum concentration of 4 × 10^6^ spores/mL in 2017 was lowered to 1 × 10^5^ spores/mL in subsequent years to better mimic natural infections. However, this inoculum concentration is still thousands of folds higher than what maize ears are normally exposed to under natural conditions. The use of such a high concentration is to examine the aflatoxin resistance of our transgenic lines under extreme conditions. Our transgenic lines have repeatedly produced significantly less aflatoxin than their controls and demonstrated the reliability of the resistance offered by HIGS. This is also the reason for the extremely higher levels of aflatoxin being detected in our KSA assays compared to those in inoculated field studies or those seen under natural infections.

Different inoculation methods have been used historically to evaluate aflatoxin resistance, such as dipping inoculation for evaluation of mature kernels under laboratory KSA conditions, spraying on silk and side-needle or pin-bar inoculation on immature kernels under field condition to assess maize lines for resistance to *A. flavus* infection or aflatoxin production. In the present study, the dipping method was used for the laboratory evaluation of mature kernels at T1 and T3 without wounding the kernels. This should allow the expression of resistance that could be associated with kernel pericarp that is circumvented by wounding when the side-needle inoculation method was used in the evaluation of immature kernels under field conditions. In addition, the side-needle inoculation method has been shown to be more effective for infection of developing maize kernels compared to the silk channel inoculation method [46,64,65,66]. Reduced aflatoxin production at an average of 84% was observed during field evaluation of the transgenic lines using the side-needle inoculation method. Similar reduction in aflatoxin production was observed (up to 87%) in the kernels of transgenic lines under KSA assays. This suggests that even in the case of seed injury or damage caused by insect vector under field condition, aflatoxin accumulation could still be significantly lower in the transgenic lines.

## 5. Conclusions

The RNAi vector was introduced into maize to target the sequence homologous to the *A. flavus* alkaline protease (*alk*) gene for silencing enhanced aflatoxin resistance in transgenic maize lines. The resistance to aflatoxin mediated by HIGS varies among the transformation events and the hybrids developed and was consistently observed over multiple evaluation trials across multiple transformation events. This research demonstrates that HIGS has great potential in developing aflatoxin resistant maize genotypes and/or improving already existing aflatoxin susceptible commercial maize germplasms. Future efforts to combine multiple key fungal infection or aflatoxin biosynthesis-related genes in a single RNAi construct may provide a more durable and potent suppression of *A. flavus* infection and subsequent aflatoxin contamination in maize.

## Figures and Tables

**Figure 1 jof-07-00904-f001:**
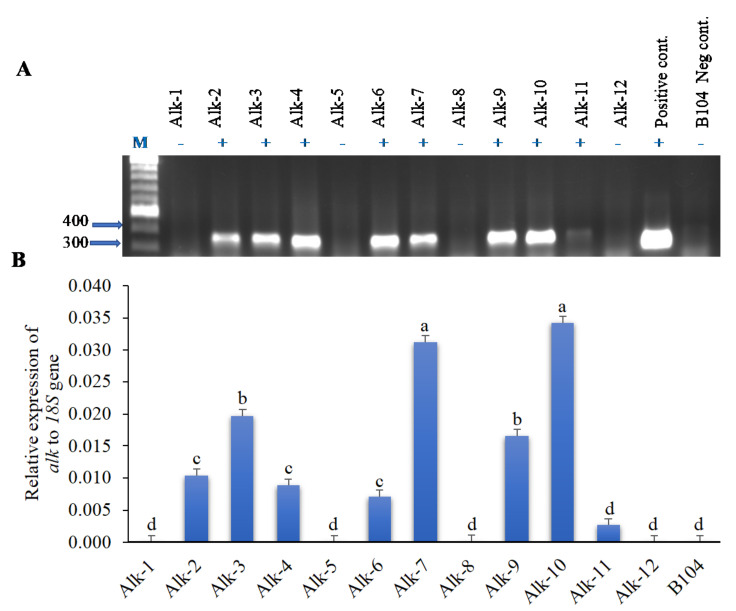
The presence (or absence) and the level of expression of the alkaline protease gene in T0 transgenic leaf tissues. (**A**) PCR confirmation of the presence (+) or absence (−) of the alkaline protease gene (*alk*) in pTF102-Alk-RNAi vector transformed T0 leaf tissues. pTF102-Alk-RNAi plasmid DNA was used as a positive control, and the genomic DNA from maize line B104 was used as a negative (−) control. (**B**) Expression of the *alk* transgene in the T0 leaf tissue of various transformation events relative to the maize 18S rRNA using real time PCR. Alk-1, Alk-5, Alk-8 and Alk-12 are negative for the transgene. Events labelled with the same letters are not significantly different at *p* ≤ 0.05.

**Figure 2 jof-07-00904-f002:**
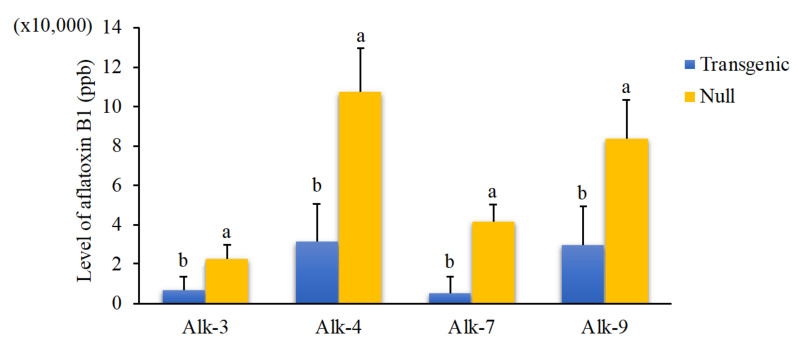
Aflatoxin production in the Alk-RNAi homozygous transgenic and null kernels of Alk-3, Alk-4 Alk-7 and Alk-9 events at T3 generation under in vitro Kernel Screening Assay (KSA) conditions. Analysis was carried out using 20 replicates (one kernel/replicate) each of homozygous and null controls. Bars with different letters are significantly different at *p* ≤ 0.05. *A. flavus* inoculum concentration of 4 × 10^6^ conidia/mL was used for this experiment.

**Figure 3 jof-07-00904-f003:**
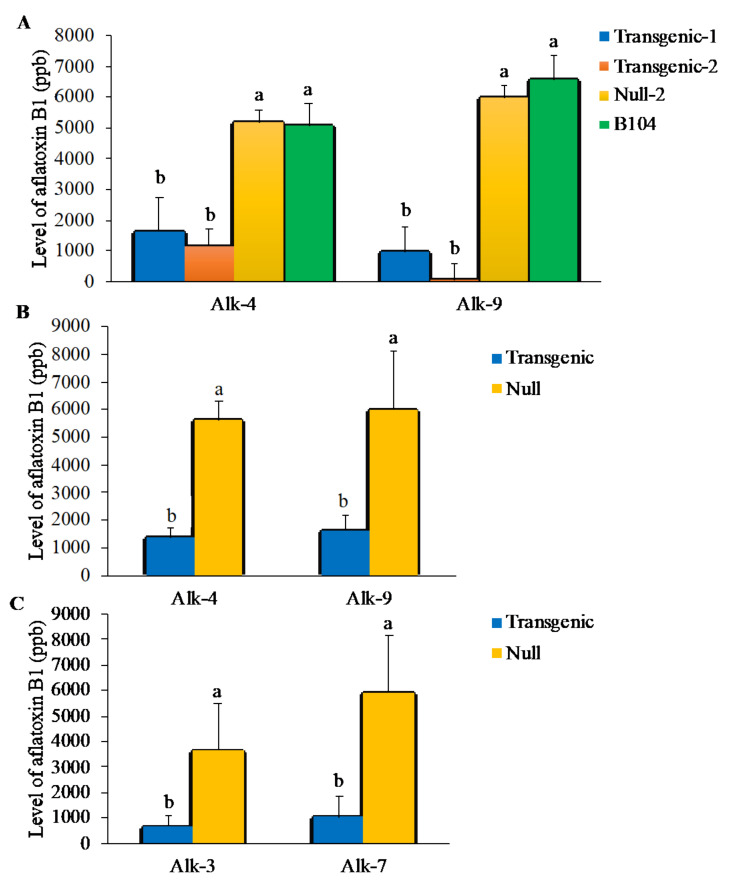
Evaluation of homozygous Alk transgenic lines under field inoculation conditions. (**A**) Aflatoxin production in T5 generation mature kernels of transgenic Alk lines and their null controls under field inoculation conditions. (**B**) Aflatoxin production in the T6 generation mature kernels of transgenic Alk-4 and Alk-9 lines and their null controls under field inoculation conditions. (**C**) Aflatoxin production in the T4 generation mature kernels of transgenic Alk-3 and Alk-7 lines and their null controls under field inoculation conditions. The inoculum concentration of 1 × 10^5^ conidia/mL was used for these experiments. Bars with different letters are significantly different at *p* ≤ 0.05. Analysis was carried out using 10–14 ears with 3–4 samples per ear for each of homozygous, null and B104 lines. Transgenic represents the kernels from ears that contain *alk*. Null refers to kernels from ears of the same transformation events without the *alk* gene.

**Figure 4 jof-07-00904-f004:**
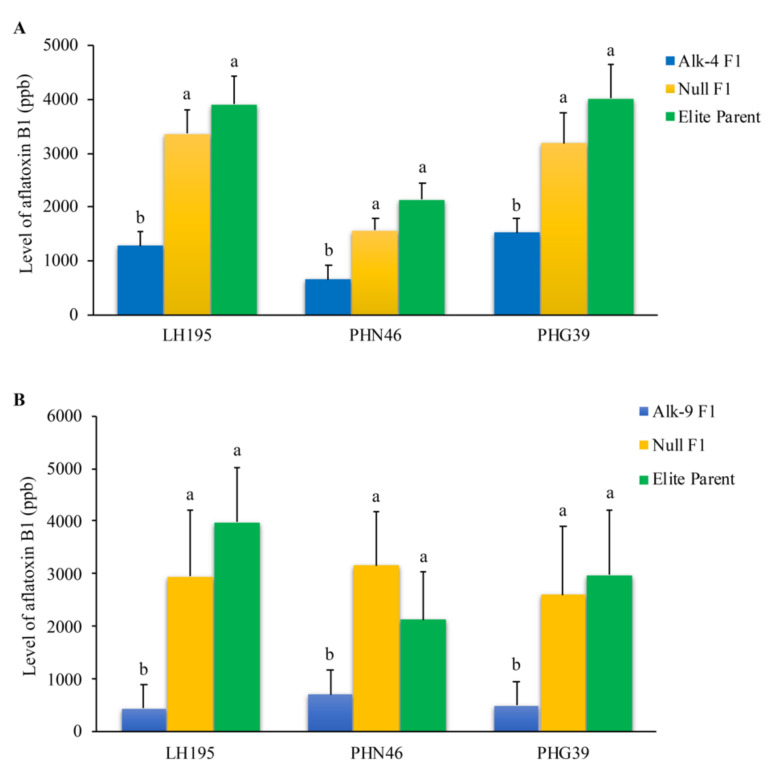
Aflatoxin production in mature kernels of the F1 crosses between Alk-4 (**A**) and Alk-9 (**B**) transgenic and null plants with three elite inbred lines (LH195, PHN46 and PHG39) under field inoculation conditions. Significant differences were observed in aflatoxin accumulation (*p* < 0.006 to *p* < 0.045) of F1 crosses between Alk-4/Alk-9 homozygous transgenic line and elite inbred lines compared with control F1 crosses between Alk-4/Alk-9 null and elite inbred lines as well as self-pollinated elite lines. Inoculum concentration of 1 × 10^5^ conidia/mL was used for this experiment. Four different points were inoculated on each ear. Lower half of each ear with the four inoculation points was ground and 3 subsamples per ear were used for aflatoxin extraction. This analysis was carried out using 27–45 sub-samples from 9–15 ears for each cross.

**Figure 5 jof-07-00904-f005:**
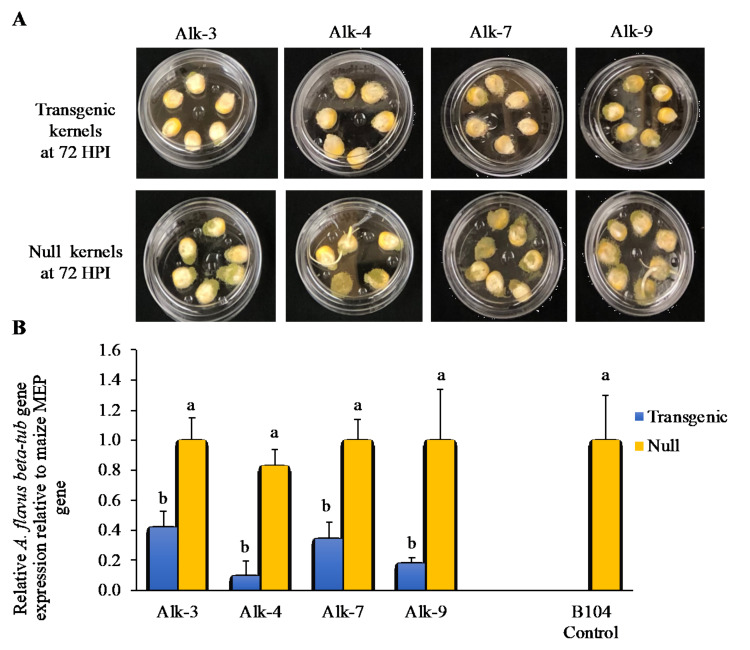
*A. flavus* growth inside inoculated Alk-RNAi and null maize kernels. (**A**) Transgenic maize kernels (top) and the null kernels (bottom) from the respective events 72 h after *A. flavus* inoculation. (**B**) The levels of *βeta-tubulin* DNA in the inoculated transgenic and null kernels relative to the DNA of maize membrane protein (MEP) gene. Significantly lower *A. flavus* biomass (based on the level of *A. flavus β-tub* DNA) were detected in the transgenic compared to the null kernels in all four events. Each of the bars represents the mean value ± standard deviation of three replicates per sample. Bars with different letters are significantly different (*p* < 0.05).

**Figure 6 jof-07-00904-f006:**
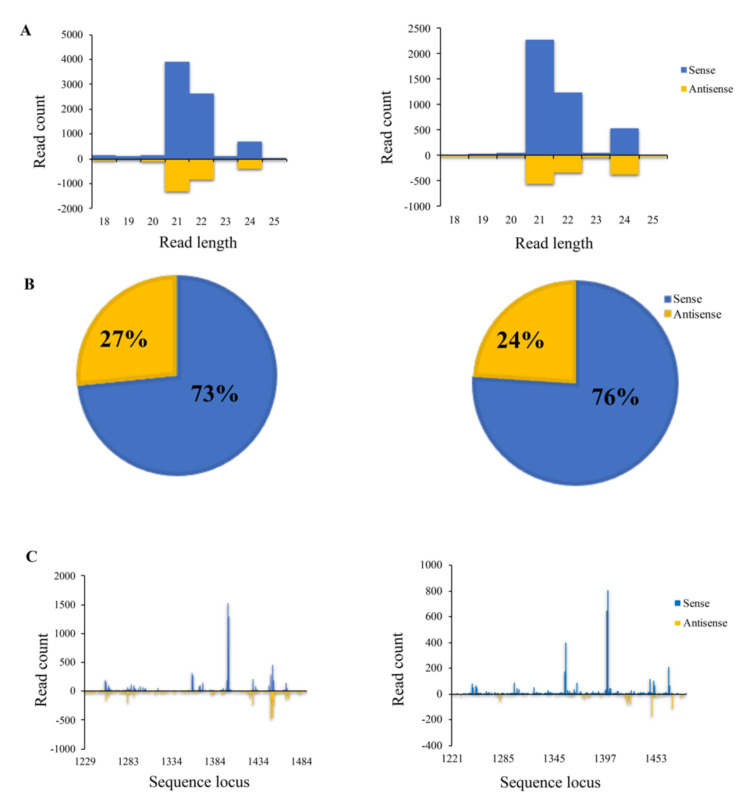
Distribution and frequency of small RNAs aligned to *alk* target gene in transgenic maize immature kernel and leaves tissues. (**A**) High levels of 21, 22 and 24 bp *alk*-specific small RNA were detected in transgenic leaves (left) and kernels (right) of the Alk-4 maize. (**B**) Both anti-sense and sense small RNA were detected in the transgenic tissues. A higher percentage of the *alk*-specific sense strand than that of the anti-sense strand was detected. (**C**) The region where the highest *alk*-specific read counts were found is between the 290 bp, which coincides with the size of the *alk* gene. The horizontal axis represents the relative position along the *alk* gene, and the vertical axis represents the number of *alk*-small RNAs reads mapped to *alk*.

**Table 1 jof-07-00904-t001:** Information on the self-pollination of Alk lines from 2017 to 2019 and crosses produced with three elite inbred lines in 2018 and 2019.

Year	Lines	# of Seeds	# of Transplant	# of Transgenic	# of Plants Pollinated	# of Ears Inoculated ^$^	Total # of Ears Harvested
2017 selfing (T3 to T4)	Alk-4	150	103	67	45	12	43
	Null-4	70	58	N/A	47	12	47
	Alk-9	148	122	46	36	14	36
	Null-9	70	58	N/A	45	10	39
	B104-13	100	61	N/A	54	7	23
2018 selfing (T4 to T5 and F1 crossing	Alk-4	240	180	180	95	23	73
	Null-4	200	190	N/A	61	15	61
	Alk-9	120	80	80	77	26	73
	Null-9	120	100	N/A	60	13	42
	B104-9	100	87	N/A	57	14	44
	LH195	206	169	N/A	33 *	29	30
	PHN46	197	174	N/A	37 *	29	32
	PHG39	210	180	N/A	31 *	24	29
	Alk-4		50				
	Null-4		50				
2019 selfing T5 to T6 (Alk-4, 9) and T3 to T4 (alk-3, 7) and crossing	Alk-4	75	52	52	≥40	15	35
	Null-4	75	55	N/A	≥40	15	35
	Alk-9	75	45	45	≥40	15	36
	Null-9	75	48	N/A	≥40	15	38
	Alk-3	160	150	35	35	12	30
	Null-3	70	60	N/A	59	12	57
	Alk-7	160	150	37	37	15	31
	Null-7	70	57	N/A	50	12	42
	LH195	150	131	N/A	35 *	25	33
	PHN46	150	140	N/A	38 *	32	35
	PHG39	150	137	N/A	36 *	24	32
	Alk-9	60	54				
	Null-9	60	56				

^$^ Number of ears inoculated at 2 weeks after pollination. N/A: Not Applicable. *Among these number of inbred plants, 1/3 were crossed with homozygous Alk-4 or Alk-9, 1/3 of them were with null Alk-4 or Alk-9, and the remaining 1/3 were self-pollinated as control.

**Table 2 jof-07-00904-t002:** *Alk* gene copy number analysis based on genotyping of seedlings developed from self-pollinated T3 generation ears and chi-square analysis.

Line	# of Seedlings	Seedlings Containing Transgene (O/E) *	Seedlings without Transgene (O/E)	X^2^	# of Integration
Alk-3	100	74/75	26/25	0.05	1
Alk-4	87	64/65	23/22	0.1	1
Alk-7	96	77/72	19/24	1.4	1
Alk-9	89	70/67	19/22	0.5	1

* The observed (O) and expected (E) number of seedlings with or without the transgene were determined through genotyping and calculated based on an expected ratio of 3:1 for single integration, respectively. Number of transgene integrations was estimated based on the whether the calculated chi-square Χ^2^ value exceeds the critical value of 3.841 to reject or accept the null hypothesis of being one or more integrations.

**Table 3 jof-07-00904-t003:** Determination of transgene copy numbers of T0 and T4 leaf tissues using droplet digital PCR.

Event	*Alk* Copy/µL	*Adh1* Copy/µL	*Alk*/*Adh1*	Copy Number *
Alk-3 (T0)	330	302	1.09	1 (hemi)
Alk-4 (T0)	563	603	0.93	1(hemi)
Alk-6 (T0)	238	245	0.97	1 (hemi)
Alk-7 (T0)	85.9	48.6	1.76	2 (hemi)
Alk-9 (T0)	540	249	2.17	2 (hemi)
Alk-10 (T0)	593	274	2.16	2 (hemi)
Alk-11 (T0)	187	181	1.03	1 (hemi)
Alk-3 (T4)	342	153.7	2.23	1 (homo)
Alk-4 (T4)	529	290	1.82	1 (homo)
Alk-7 (T4)	144.4	36.7	4.20	2 (homo)
Alk-9 (T4)	256.8	54.9	4.68	2 (homo)

* The target gene copy number was calculated based on the ratio between number of molecules of target *alk* gene and maize single copy reference alcohol dehydrogenase gene (*adh1*) in the genomic DNA samples. The primers for *alk* gene used in the ddPCR has one target in each of two *alk* arms in the construct.

**Table 4 jof-07-00904-t004:** Phenotypic assessment of homozygous Alk-RNAi lines in comparison to the controls.

Line T4 Plants	Mean Plant Height (cm)	Days to Tasseling	Days to Silking	Mean Cob Length (cm)	Mean 100 Seed Weight (g)
Alk-3 (Transgenic)	138.2ab	69.0b	72.0b	13.5d	23.4c *
Alk-3 (Null)	136.8b	68.1bc	71.0bc	13.1d	21.5d *
Alk-4 (Transgenic)	138.9ab	67.0bc	70.0c	13.3cd	22.1d *
Alk-4 (Null)	140.2ab	67.2bc	68.2d	13.7cd	22.1d *
Alk-7 (Transgenic)	137.9ab	66.1cd	68.9cd	15.1a *	24.1bc
Alk-7 (Null)	141.2ab	68.0bc	71.0bc	15.0a *	25.4a
Alk-9 (Transgenic)	132.3c *	71.0a *	74.0a *	13.3d	23.8c *
Alk-9 (Null)	142.0a	67.0c	70.0c	14.6bc	24.3ab
B104	139.4ab	68.0bc	70.0c	14.3bd	24.8ab
** Std. dev.	2.9	1.5	1.7	0.7	1.3

* Transformation event that showed significant differences in plant height, days to tasseling/silking, ear length and kernel weight when compared to its null and the B104 line at *p* < 0.05. The different letters represent lines that are significantly different from the B104 line at *p* < 0.05. ** Standard deviation values for each phenotypic trait.

**Table 5 jof-07-00904-t005:** Number of small RNA reads in leaf and immature kernels tissues of transgenic and non-transgenic maize lines.

Tissue Type	Event	Total Read	Reads Aligned to *A. flavus*	Reads Aligned to *Alk*
Leaf	Alk-4 (Homo)	36,025,471	13,699	9574
	B104 (Null)	60,780,742	6238	7
kernels	Alk-4 (Homo)	30,112,812	52,476	6606
	Alk-4 (Null)	35,988,183	64,291	8

## Data Availability

The datasets generated in this study are available in the NCBI SRA database under the accession number: PRJNA759324.

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
