# Peer review of "Resistance to Aflatoxin Accumulation in Maize Mediated by Host-Induced Silencing of the *Aspergillus flavus* Alkaline Protease (*alk*) Gene"

_jof, 2021, doi:10.3390/jof7110904_

Round 1

Reviewer 1 Report

The article presented by Omolehin and co-workers shows an RNAi system to inhibit the growth of Aspergillus flavus on maize plants and, consequently, reduce aflatoxin contamination. The RNAi system was based on the alk gene, which codes for an alkaline protease identified in A. flavus. While the methodology is not new, the article is well written and well presented. The experimental procedure has been very well explained and all the necessary technical steps seem to be correct. The final conclusions are experimentally supported and I have not found any inconsistencies in the presented work. I have only minor suggestions.

Line 79: “…their resistance genes [45] .” Please remove the space before the dot.

Line 126: You wrote “This is the first study in targeting A. flavus alkaline protease gene…”. This is not wrong, but it is a bit overselling. As I said, even if the methodology isn't new, your findings are still valuable.

Figure 1 can go in the supplementing…

Author Response

Responses to reviewers’ comments to JOF-1415096

Reviewer 1:

Comment: The article presented by Omolehin and co-workers shows an RNAi system to inhibit the growth of Aspergillus flavus on maize plants and, consequently, reduce aflatoxin contamination. The RNAi system was based on alk gene, which codes for an alkaline protease identified in A. flavus. While the methodology is not new, the article is well written and well presented. The experimental procedure has been very well explained and all the necessary technical steps seem to be correct. The final conclusions are experimentally supported, and I have not found any inconsistencies in the presented work. I have only minor suggestions.

Response: We appreciate the reviewer’s comments.

Other minor corrections suggested are addressed below:

Comment: Line 79: “…their resistance genes [45] ” Please remove the space before the dot.

Response: The suggested change has been adopted.

Comment: Line 126: You wrote “This is the first study in targeting A. flavus alkaline protease gene…”. This is not wrong, but it is a bit overselling. As I said, even if the methodology isn’t new, your findings are still valuable.

Response: We appreciate the reviewer’s comment and have reworded our statement to truly reflect the fact and to avoid giving the “overselling” impression.

Comment: Figure 1 can go in supplementing…

Response: Figure 1 has been moved to be the supplementary section as Figure S2 and consequently, changes have been made to the numbering of the remaining figures.

Reviewer 2 Report

Respected colleagues,

Congratulations on well presented interesting and demanding work. Please see a few improvement suggestions in comments in the manuscript file.

Regards

Author Response

Responses to reviewers’ comments to JOF-1415096

Reviewer 2:

Comment: Congratulations on well-presented interesting and demanding work. Please see a few improvement suggestions in the comments in the manuscript file.

Response: We appreciate the reviewer’s comments.

Other minor corrections suggested are addressed below:

Comment: Line 29: This (Introduction) section is too long. Make it shorter, move deleted sections to discussion

Response: We have significantly shortened the introduction section as suggested by deleting some of the information, such as lines 33-34, 39-42, 49-51, 54-65 (the paragraph on maize breeding effort and on transgenic effect), 69-75, 85-89, 97-99, and 100-101 of the original manuscript. The removed information was not incorporated into discussion due to the lack of a close connection to the information in the discussion.

Comment: Line 135: Indicate primer sequence here

Response: The alk primer sequences have been inserted into the body of the text as suggested. The other primer sequences referred to in this section are presented in Table S1.

Comment: Figure 5: Indicate top (transgenic kernels) and bottom (null kernels) in the photo also.

Response: Thank you for catching our oversight of the very important part of this illustration. The labels on Figure 5 of this revised manuscript have been changed to reflect null rather than transgenic kernels in the bottom section.